# Model of Chronoamperometric Response towards Glucose Sensing by Arrays of Gold Nanostructures Obtained by Laser, Thermal and Wet Processes

**DOI:** 10.3390/nano13071163

**Published:** 2023-03-24

**Authors:** Antonino Scandurra, Valentina Iacono, Stefano Boscarino, Silvia Scalese, Maria Grazia Grimaldi, Francesco Ruffino

**Affiliations:** 1Department of Physics and Astronomy “Ettore Majorana”, University of Catania, via Santa Sofia 64, 95123 Catania, Italy; valentina.iacono@dfa.unict.it (V.I.); stefano.boscarino@dfa.unict.it (S.B.); mariagrazia.grimaldi@ct.infn.it (M.G.G.); francesco.ruffino@ct.infn.it (F.R.); 2Institute for Microelectronics and Microsystems of National Research Council of Italy (CNR-IMM, Catania University Unit), via Santa Sofia 64, 95123 Catania, Italy; 3Research Unit of the University of Catania, National Interuniversity Consortium of Materials Science and Technology (INSTM-UdR of Catania), via S. Sofia 64, 95125 Catania, Italy; 4Institute for Microelectronics and Microsystems of National Research Council of Italy (CNR-IMM), Ottava Strada, 5 (Zona Industriale), 95121 Catania, Italy; silvia.scalese@imm.cnr.it

**Keywords:** laser processing, dewetting, gold nanostructures, glucose sensing, convergent diffusion, reaction kinetics

## Abstract

Non-enzymatic electrochemical glucose sensors are of great importance in biomedical applications, for the realization of portable diabetic testing kits and continuous glucose monitoring systems. Nanostructured materials show a number of advantages in the applications of analytical electrochemistry, compared to macroscopic electrodes, such as great sensitivity and little dependence on analyte diffusion close to the electrode–solution interface. Obtaining electrodes based on nanomaterials without using expensive lithographic techniques represents a great added value. In this paper, we modeled the chronoamperometric response towards glucose determination by four electrodes consisting of nanostructured gold onto graphene paper (GP). The nanostructures were obtained by electrochemical etch, thermal and laser processes of thin gold layer. We addressed experiments obtaining different size and shape of gold nanostructures. Electrodes have been characterized by field emission scanning electron microscopy (FE-SEM), X-ray photoelectron spectroscopy (XPS), cyclic voltammetry, and chronoamperometry. We modeled the current-time response at the potential corresponding to two-electrons oxidation process of glucose by the different nanostructured gold systems. The finest nanostructures of 10–200 nm were obtained by laser dewetting of 17 nm thin and 300 °C thermal dewetting of 8 nm thin gold layers, and they show that semi-infinite linear diffusion mechanism predominates over radial diffusion. Electrochemical etching and 17 nm thin gold layer dewetted at 400 °C consist of larger gold islands up to 1 μm. In the latter case, the current-time curves can be fitted by a two-phase exponential decay function that relies on the mixed second-order formation of adsorbed glucose intermediate followed by its first-order decay to gluconolactone.

## 1. Introduction

Nanomaterials are among the most important topics of scientific and technological research in the 21st century [1]. In fact, the last decades have seen all the technological fields, connected to the challenges of human society, exploit the potential offered by nanomaterials [2]. This is happening since nanomaterials, which have at least one dimension on the nanoscale, exhibit significant unconventional physical and chemical properties that are very different from those of the corresponding bulk materials [3,4,5,6]. A field that takes advantage of the use of nanomaterials in a significant way is that of electrochemistry and, in particular, of electrochemical sensors [7].

Electrochemical sensors exploit two main advantages offered by nanomaterials. The first is given by the electro-catalytic properties of nanomaterials towards various oxidation or reduction reactions, showing enzyme-like catalytic properties towards biological analytes, which has earned them the term nano-zymes [8,9]. The second advantage derives from the fact that an electrode fabricated using an array of small active nanostructures, suitably spaced by inactive substrate material, exhibits the change of the diffusion mechanism of the analyte from semi-infinite linear to convergent [10]. This condition favors a high mass transport rate at the electrode surface, thus extending the saturation limit of the active sites [11].

The success of nanomaterials in the design and production of electrochemical sensors depends intimately on the development of versatile, low-cost, simple, and high-throughput nano-fabrication and nano-patterning techniques [12,13]. In recent years, laser-matter interaction opened up new scenarios for the fabrication of nanomaterials, in particular, by controlling the laser process parameters. Thus, a wide variety of nanostructures have been obtained by exploiting this methodology [14,15,16]. Recently, Khorasani et al., reported a review on the process and production features offered by laser subtractive and powder bed fusion [17]. Furthermore, we demonstrate the applicability in electrochemical sensing of a gold thin layer patterned by thermal or laser dewetting processes [18,19]. Gold nanostructures present a significant electro-catalytic activity towards glucose oxidation and other non-enzymatic process [20]. Based on this property, amperometric non-enzymatic glucose sensors have been successfully reported in the literature [21,22]. This type of sensor is of paramount relevance due to the possibility of improving the quality of life of patients with diabetes through continuous blood glucose monitoring systems [23].

The amperometric mode of operation of the electrochemical sensor offers further advantages over other approaches like voltammetric or impedentiometric, such as, for example, the possibility of having a current response proportional to the initial concentration of the analyte [24]. However, some complications arise since the current response depends not only on the reaction kinetics of oxidation or reduction of the analyte:Red ⇆ Ox + e^−^(1)
but also on the morphology of the electroactive nano-material and on the mass transport of the analyte towards the electrode surface. Mass transport of the analyte toward the electrode surface is often under the control of its diffusion [24]. In turn, the diffusion mechanism of the analyte could change from semi-infinite planar to convergent. The diffusion mechanisms may be defined by the size of the nanostructures [25]. Thus, the description of the system becomes complex and only in a few canonical cases has it been possible to model the current-time response, as in the case of a macro-electrode using the Cottrel equation [26], finite micro-discs [27,28], or spheres [29]. Accordingly, the study of the current response as a function of nanostructure morphology represents an important issue both in fundament research as well as in the sensor design [30].

In this paper, we describe the electrochemical responses toward glucose determination in an alkaline solution by four nanostructured gold systems obtained by wet, thermal, and laser processes. The time-current dependence of the two-electron oxidation process of glucose was modeled for the considered gold nanostructures. Furthermore, we investigated whether the transient current response can be modeled by planar or convergent diffusion-limiting mechanisms or whether there are noticeable kinetic contributions that need to be taken into account.

## 2. Materials and Methods

### 2.1. Materials and Electrode Fabrication

Graphene paper (GP) 240 μm thick, sodium hydroxide, and D (+) glucose both 99.99% of purity were purchased from Sigma Aldrich (Milan, Italy). The solution of glucose in NaOH 0.1 M was prepared by de-ionized water treated in a MilliQ™ system. The typical concentration of carbon contaminants was at ppb (part per billion) level and the electrical resistivity was 18.2 MΩ × cm, respectively. Emitech K550X sputter coater (Quorum, Ringmer, East Sussex. UK) was used for the gold layers deposition. Electrodes were fabricated by gold deposition onto 1 cm × 1 cm square area of 1 cm × 3 cm pieces of graphene paper, at conditions of 50 mA for 45 and 90 s. These conditions correspond to gold layers 8 nm and 17 nm thin, respectively. The layer homogeneity and thickness were analyzed by Rutherford Backscattering Spectrometry (RBS) (High Voltage Engineering Europa B.V., Amersfoort, the Netherlands), simulating the spectra by using XRump software (genplot, www.genplot.com (accessed on 1 March 2023)) [31]. The not-metalized portion of the electrode was isolated by adhesive tape.

Four gold nanostructures were prepared in this study: (a) nanoporous by 17 nm thin gold layer electrochemically etched in NaOH 0.1 M, by sweeping the potential between −0.5 to +1 V vs. SCE for 5 cycles; (b) and (c) dewetting by thermal annealing of the 8 nm and 17 nm gold layers at 300 and 400 °C, respectively, in N_2_ atmosphere, in a Carbolite Gero oven (Verder Scientific, GmbH & Co. Retsch-Allee 1-5 42781 Haan, Germany); (c) dewetting of 17 nm thin gold layer by nanosecond pulsed laser irradiation. Nd: yttrium aluminum garnet YAG laser, operating at 532 nm of wavelength was used (Quanta-ray PRO-Series pulsed Nd:YAG equipment, Spectra Physics, 1565 Barber Lane Milpitas, CA 95035, USA). The conditions of the irradiation process were 10 ns pulse duration, fluence of 0.5 Jcm^−2^, in air. In the used experimental conditions, the typical spot size of the laser beam was 2 mm. To fully irradiate an area of 1 cm^2^, several nearby spots were made, partially overlapping each other. In all systems, the electrode working area was 1 cm^2^. The employed experimental conditions of thermal and laser dewetting were optimized experimentally in order to obtain reproducible results.

### 2.2. Instrumental Characterization

The morphology of nanostructured gold systems was obtained by Gemini 152 Carl Zeiss Supra 35 FE-SEM (Jena, Germany), operating with the detector in the in-lens mode. Typical instrumental parameters were working distance of 3 mm; a beam acceleration potential of 5 kV, and an aperture size of 30 μm.

The electronic structure of the nanostructured gold surface was investigated by XPS. PHI 5000 Versa Probe II system ULVAC-PHI, Inc. (2500 Hagisono, Chigasaki, Kanagawa, Japan) was used for the characterization. The spectra were excited by monochromatized Al Kα X-ray radiation. Linear background subtraction was used to analyze the photoelectron peaks. The Au 4f_7/2_ peak of untreated gold reference, which is centered at 84 eV of binding energy [32,33], was used for the binding energy scale calibration. Electrochemical measurements were performed in air at 22 °C by a potentiostat VersaSTAT 4 (Princeton, Oak Ridge, TN, USA). In each measurement, 30 mL of fresh solutions of glucose at various concentrations in NaOH 0.1 M were used. The potential of the working electrodes was referenced to the saturated calomel electrode (SCE). Platinum wire was used as a counter. The electrochemical characterization of gold nanostructures was performed by cyclic voltammetry (CV) at a scan rate of 20 mVs^−1^ and amperometric measurements.

## 3. Results

### 3.1. Morphology and Electronic Structure

Figure 1a,b show the FE-SEM morphology of the nanoporous gold, taken at two magnifications, obtained by five cycles in NaOH 0.1 M between −0.5 and 1 V. The surface presents numerous voids that are formed by preferential gold etching along the edges of the graphene planes. The etched regions appear in the FE-SEM picture as a dark spot and have sizes ranging from a few to several hundreds of nanometers. Figure 1c,d show, at two different magnifications, the morphology of the 8 nm thin gold layer dewetted at 300 °C. The gold layer after dewetting forms narrow spaced islands having a size between 10 and 200 nm. The gold islands are spaced by an underlying substrate of GP (shown by dark regions in the FE-SEM pictures) where the gold has been dewetted. The spacing between gold islands is in the range of 10–200 nm. The pictures of Figure 1e,f show the FE-SEM morphology of the 17 nm thin gold layer after thermal dewetting at 400 °C. Gold islands are larger than in the case of the 8 nm film, with sizes ranging from 20 nm to about 1 μm. Furthermore, the spacing between gold islands ranges between 20 nm to about 1 μm. Figure 1g,h show the FE-SEM morphology of the laser-irradiated gold nanostructure. The surface is formed by exfoliated graphene paper. Exfoliation operated by the laser pulse produces thin nanowalls formed by a few layers of graphene. Moreover, almost spherical-shaped gold nanostructures are formed on the graphene multilayers because of the thin film melting operated by the laser pulse. The average size of the gold nano-spheres is around 20 nm.

Electronic structure was studied by XPS. Figure 2 shows the Au 4f spectral regions of the four gold nanostructures. The spectra of nanoporous, 8 nm dewetted at 300 °C and 17 nm dewetted at 400 °C gold (Figure 2a–c) show the Au 4f_7/2,5/2_ spin-orbit components centered at binding energy of 84 and 87.7 eV (3.7 eV spin-orbit coupling), respectively, thus indicating the presence of Au^0^ states [32,33]. Instead, the spectrum of laser-dewetted gold (Figure 2d) shows the Au 4f_7/2,5/2_ spin-orbit components centered at 84.35 and 88.05 eV (3.7 eV spin-orbit coupling), respectively. The observed shift of the binding energies can be assigned to the presence of Au(I) specie [34], which are formed by laser interaction with the gold layer, characterized by a high-temperature process.

### 3.2. Electrochemical Characterization

Recently, some authors reported that glucose content in interstitial or other biological fluids is not readily in equilibrium with the glucose content in blood, but there is a time-lag to reach the equilibrium [35]. Thus, some non-invasive glucose monitoring systems pursue the direct sampling blood through the subcutaneous tissues. Consequently, even if we have demonstrated that our gold nanostructures have very competitive detection limits in the range of a few μM of glucose, in this work we have focused our attention on typical glucose concentration in blood that covers the normal but also pathological health conditions, that is, in the range from 2.5 to 10 mM.

Figure 3a–d show the cyclic voltammograms of glucose at the concentration of 2.5, 5, 7.5, and 10 mM, obtained by (a) gold nanoporous, (b) 8 nm dewetted at 300 °C, (c) 17 nm dewetted at 400 °C, and (d) 17 nm dewetted by laser pulse at 0.5 J cm^−2^ of fluence, respectively. The voltammograms show four main features, whose relative intensity depends on the type of gold nanostructure. In particular, in the forward scan ongoing from the potential of −0.5 to 1 V vs. SCE, the peak 1 at about −0.15 V is assigned to glucose adsorption on the gold surface by dehydrogenation [36,37].
C_6_H_12_O_6_ + OH^−^ + Au → Au(C_6_H_11_O_6_)_ads_ + H_2_O + e^−^(2)

The peak 2 is assigned to the electro-sorption of OH^-^ on the gold surface, and the formation of gold hydroxide intermediate complex:Au + OH^−^ ⇆ [Au(OH)_ads_]^−^(3)

At this step the adsorbed glucose is oxidized to gluconic acid:Au(C_6_H_11_O_6_)_ads_ + [Au(OH)_ads_]^−^ → [Au--O-(C_6_H_10_O_6_)_ads_]^−^ + Au + H_2_O(4)
[Au--O-(C_6_H_10_O_6_)_ads_]^−^ → AuOH + C_6_H_10_O_6_ + e^−^(5)

Further increasing the positive potential to about 0.3 V, peak 3 appears, which is assigned to the oxidation of the gold hydroxide intermediate to gold oxide (Equation (6)).
2 [Au(OH)_ads_^]−^ ⇆ Au_2_O + H_2_O + 2e^−^(6)

At this step the glucose oxidation is inhibited. Interesting, the voltammogram of the laser dewetted sample, since the surface is already formed by gold oxide, shows the lowest intensity of the peak 3. In the backward scan, glucose is oxidized by a two-electrons mechanism, producing the intense peak 4:Au_2_O + C_6_H_12_O_6_ → C_6_H_10_O_6_ + 2 AuOH(7)

Furthermore, gold oxide is reduced back to gold hydroxide. The position and intensity of the two-electrons oxidation peak depend on the glucose concentration and on the type of gold nanostructure.

Figure 4 reports the current density of the two-electron oxidation peak as function of the peak potential vs. SCE. Current density is related to the actual potential E by the Butler–Volmer Equation:
(8)$$J={\;k}_{0}F\left\{c_{\mathrm{glucose}}e^{\left[\frac{\alpha_{a}F}{\mathrm{RT}}\left(E-E_{f}^{0}\right)\right]}\right\}$$
where, taking into account the anodic portion, J is the current density, k_0_ is the heterogeneous rate constant, c_glucose_ is the bulk concentration of glucose, α_a_ is the anodic charge transfer coefficient, and E_f_^0^ is the formal potential of the redox couple. Accordingly, the current density depends on the k_0_, α_a_, and E_f_^0^, fixing the analyte concentration c_glucose_.

The nanostructures obtained by gold 8 nm, 300 °C thermal and 17 nm, lasers dewetting show, at the same glucose concentration and current density, lower potentials than the nano-porous and 17 nm, 400 °C. The data indicate that the overpotential of glucose oxidation on these nanostructures is lower than that shown by other gold-based nanostructures, and the electro-catalytic oxidation process is favored [37].

### 3.3. Modeling of the Current-Time Curves

Nano-electrodes array shows a number of advantages in the analytical field compared to macro-electrodes, such as great sensitivity and little dependence on the diffusion of the analyte near the electrode–solution interface. Gold dewetted nanostructures can be thought of as an array of nano-electrodes. The current-time curve is obtained by applying a proper step potential to the working electrode, whereas the current as a function of time produced by the red-ox process is measured. The time dependence of the current response for a planar macro-electrode under a semi-infinite linear diffusion control mechanism can be described by the Cottrel Equation (9) [26].
(9)$$i\left(t\right)=\mathrm{nFAc}\sqrt{\frac{D}{\pi t}}$$
where i(t) is the current, n is the number of exchanged electrons, F is the Faraday constant, A is the electrode area, c is the bulk analyte concentration, D is the diffusion coefficient of the analyte, and t is the time. For a hemispherical electrode of radius r, the Cottrel equation for the limit current became:(10)$$I_{\mathrm{limit}}=2\pi \mathrm{rnFDc}$$

Conversely, the theoretical modeling of the time dependence of the current response by nano-electrodes array is not a simple task. Several authors attempted the simulation of chronoamperometric curves by a variety of analytical [27] or numerical methods [38,39,40].

In specific conditions, when the size and interspacing of electroactive nanostructures are much smaller than the diffusion length:(11)$$\bar{\delta }=\sqrt{2\mathrm{Dt}}\gg \;r$$
where $\bar{\delta }$ is the root mean square of displacement, or diffusion length, and D is the diffusion coefficient of glucose in water-based solution at room temperature (6.7 × 10^−10^ m^2^ s^−1^) [41], the convergent diffusion approaches the semi-infinite linear diffusion as depicted in the scheme of Figure 5:

Thus, Equation (9), under some circumstances, is still valid [25,42].

Figure 6 shows the chronoamperometric curves for glucose 10 mM obtained by the four gold nanostructures. The curves were acquired within a time range of 0–150 s. Taking into consideration the early stage of the transient in the current-time curves, that is, about 0.6–1 s from the potential is stepped up, the diffusion length of glucose is about 2.8 μm. Accordingly, we found that the curves referring to the 8 nm dewetted at 300 °C and laser dewetted samples (Figure 6b,d), characterized by very small gold nanostructures (≪2.8 μm), can be fitted properly by the Cottrel Equation (9). In our experimental conditions, the nano-electrode arrays behave like macro-electrodes, and they are expected to obey Cottrel’s Equation.

The terms $\mathrm{nFAc}\sqrt{\frac{D}{\pi }}$ were included in the parameter B_0_, according to Equation (12).
(12)$$B_{0}=\mathrm{nFAc}\sqrt{\frac{D}{\pi }}$$

The equation we used for the fit was (12):(13)$$I=B_{0}\sqrt{\frac{1}{t}}$$

The parameters B_0_ used for the simulations are reported in Table 1. Since n, F, and c are the same for the two curves of Figure 6b,d, the different values of the parameter B_0_ used for the 8 nm 300 °C and laser-irradiated nanostructures reflect the extension of the active area of the two samples. In particular, the parameter B_0_ is about 4 times higher for the 8 nm 300 °C with respect to the laser-irradiated nanostructure. Thus, the two nanostructures resemble a macro-electrode. This mechanism occurs because of the gold nano-spheres and nano-islands, although the existence of a convergent diffusion mechanism can be hypothesized in the proximity of the electro-active particles separated by the inactive material, the mechanism itself is modified, and the resultant is a semi-infinite linear diffusion (see Figure 5) [10,11].

In our gold nanoporous sample, the voids (vide Figure 1b) resemble active regions, where glucose is preferentially oxidized at the gold edges. The size of the voids is comparable to that of the gold islands present in the sample 17 nm dewetted at 400 °C. For these nanostructures, we observed that do not obey Cottrel’s Equation, thus the curves of Figure 6a,c cannot be fitted by Equation (13).

At a microelectrode, the chronoamperometric response can be described [42,43]
(14)$$I=4\mathrm{nFrDcf}\left(\tau \right)$$
where $f\left(\tau \right)$ is defined by Equation (15)
(15)$$f\left(\tau \right)=0.7854+0.8862\tau^{-0.5}+0.2146e^{\left(-0.7823\tau^{-0.5}\right)}$$
that is an empirical function proposed by Shoup and Szabo [28], and τ is defined as
(16)$$\tau =\frac{4\mathrm{Dt}}{r^{2}}$$
represents the squared ratio of the diffusion length to the single micro-electrode of radius r. Hernandez proposed a model for a gold micro-disk electrode array based on stationary finite micro-disks. He used a redox probe the ferrocenecarboxylic acid (FcCO_2_H, CAS # 1271-42-7), and proposed Equation (17) to model the chronoamperometric curve [44]:(17)$$I=N\;\frac{4\mathrm{nFADc}}{\pi r}\;f\left(\tau \right)$$
where N is the number of micro-disks in area A, and r is the average radius of a single micro-disk.

Our results show that Equation (17) was not useful for the simulation of the chronoamperometric curves of Figure 6a,c, indicating that, apart from the limiting mechanism of diffusion, other mechanisms must be taken into consideration in these cases. Starting from the approach proposed by Hernandez we used an equation with an analytical solution, based on a two-phase exponential decay function, to fit our current-time curves for glucose by nanoporous and 17 nm 400 °C dewetted gold nanomaterials (18):(18)$$I=N\;\frac{4\mathrm{nFADc}}{\pi r}\left[B_{1}+B_{2}e^{-\left(k_{1}t\right)}+{\;B}_{3}e^{-\left(k_{2}t\right)}\right]$$

This function has been successfully used to fit our current-time curves of Figure 6a,c. The fitting parameters are reported in Table 1. The chemical–physical model underlying the simulation with the two-phase exponential decay function relies in the kinetic of mixed second-order formation of an intermediate followed by its first-order decay processes (19) [45]:(19)$${\mathrm{Au}}_{2}O+C_{6}H_{12}O_{6}\;\overset{k_{1}}{\to }\;[\mathrm{GLU}]*\;\overset{k_{2}}{\to }\;C_{6}H_{10}O_{6}+2\;\mathrm{AuOH}$$
where [GLU]* represent an intermediate, consisting of adsorbed glucose molecule. k_1_ and k_2_ represent the heterogeneous rate constants of glucose adsorption, forming the intermediate, and intermediate decay to gluconolactone, respectively. In Table 1, the heterogeneous rate constants k_1_ and k_2_ are similar for 17 nm 400 °C and laser dewetted samples and are effective in the fit of the curve transient in the initial few seconds. In these two gold nanostructures, due to the highly developed surface; hence, the high concentration of adsorbed molecules of glucose, the rate of oxidation process is not limited by diffusion at the early stage of current-time curves. For the transient region, the current is determined by the kinetics of the overall oxidation reaction characterized by the constants k_1_ and k_2_, and, therefore, by the exponential terms present in Equation (18). For a long time, when adsorbed glucose is consumed, the limit current is determined by the diffusion limiting factor, as in the case of systems which satisfy the Cottrellian equation. The term $N\;\frac{4\mathrm{nFADc}}{\pi r}B_{1}$ represents the steady state current.

### 3.4. Glucose Determination in Amperometric Mode

Figure 7 shows the calibration curves obtained by plotting the steady state current density measured at the potentials of the two-electron oxidation peak (vide Figure 4) as a function of glucose concentration. The calibration curves were fitted by a linear regression model by the fitting parameters reported in Table 2. For each glucose concentration, the difference in current densities reflects the different extents of the active area of gold present in the various systems. All the gold nanostructures show good linear response in the concentration interval studied. Selectivity and specificity tests of the nanostructures were done by ascorbic and uric acids. The results of these tests as well as the analytical applications of the gold nanostructures were described in previous works [19,21].

## 4. Conclusions

In this paper, we modeled the current-time dependence of the chronoamperometric curves at the potential of the two electrons oxidation peak of glucose in an alkaline solution. Four different gold nanostructures have been fabricated by wet, thermal, and laser processes, characterized by different sizes and interspacing. The finest gold nanostructures show a behavior that can be described by the Cottrel equation. Both the transient and the steady state current are limited by glucose diffusion. Conversely, the largest gold nanostructures show a transient region, where the current is determined by the kinetics of the overall oxidation reaction characterized by the constants k_1_ and k_2_, of the mixed second-order formation of the intermediate of adsorbed glucose followed by its first-order decay to gluconolactone. The mechanisms were well modeled by a two-phase exponential decay function. At times beyond the transient, when adsorbed glucose is consumed, the steady state current is determined by the diffusion limiting factor, as in the case of systems that satisfy the Cottrellian equation. The work we have presented shows how by simulating the current-time response, it is possible to obtain information both on the diffusion model, whether convergent or planar and also on the mechanisms and kinetics of the electrochemical reactions taking place at the electrode–solution interface.

## Figures and Tables

**Figure 1 nanomaterials-13-01163-f001:**
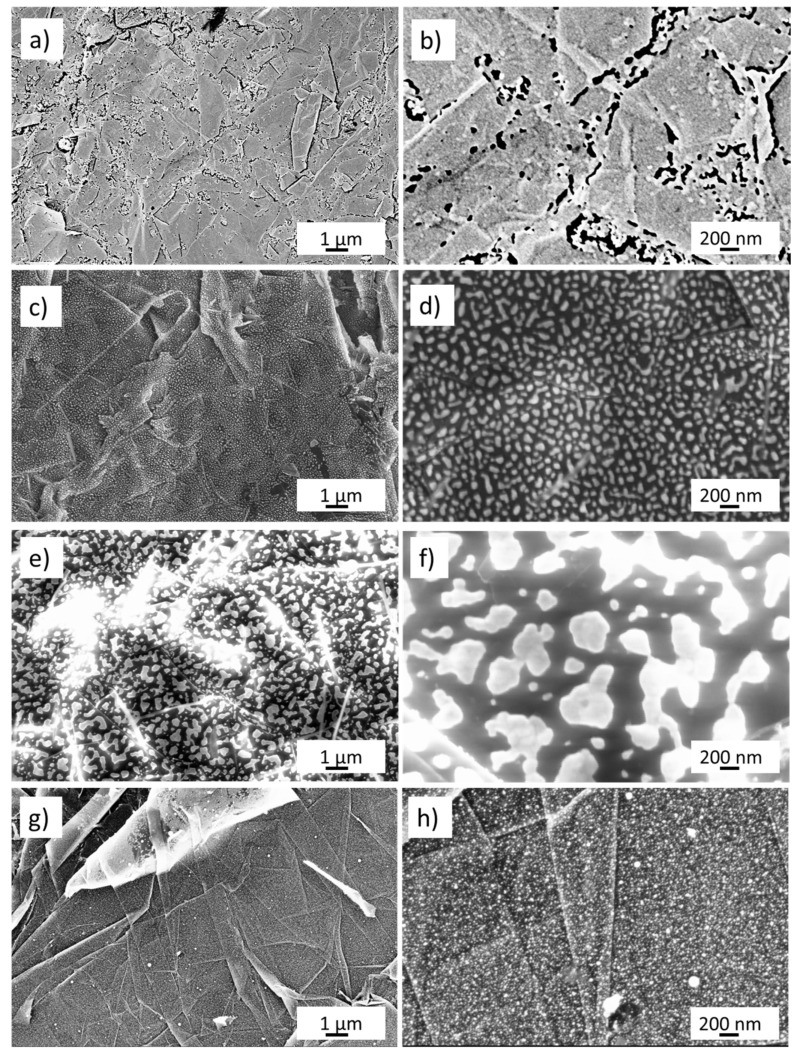
Field emission scanning electron microscopy pictures of (**a**,**b**) gold nanoporous obtained by five cycles of scanning potential between −0.5 and 1 V in NaOH 0.1 M; (**c**,**d**) gold layer 8 nm thin dewetted at 300 °C; (**e**,**f**) gold layer 17 nm dewetted at 400 °C; (**g**,**h**) gold layer 17 nm dewetted by laser at 0.5 J cm^−2^ fluence.

**Figure 2 nanomaterials-13-01163-f002:**
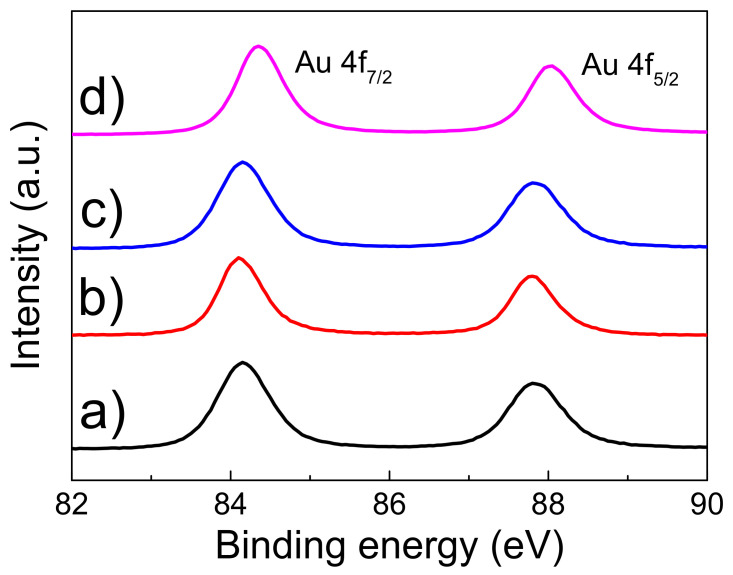
Au 4f XPS spectra of: (**a**) gold nano-porous; (**b**) 8 nm 300 °C; (**c**) 17 nm 400 °C; (**d**) laser dewetted at fluence of 0.5 J cm^−2^.

**Figure 3 nanomaterials-13-01163-f003:**
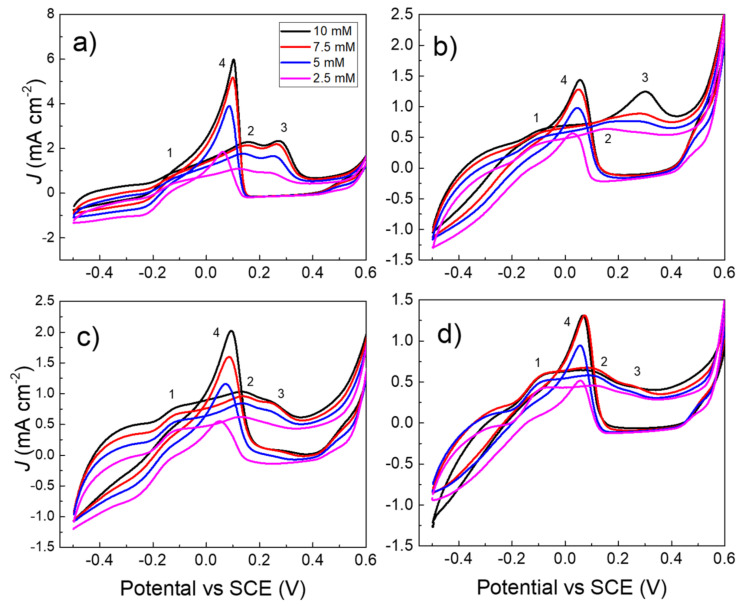
Cyclic voltammograms of glucose 2.5, 5, 7.5, and 10 mM obtained by: (**a**) gold nanoporous; (**b**) 8 nm 300 °C; (**c**) 17 nm 400 °C; (**d**) laser dewetted at fluence of 0.5 J cm^−2^. The 1–4 numbers refer to the main electrochemical processes. Condition: scan rate 20 mV s^−1^; supporting electrolyte: NaOH 0.1 M.

**Figure 4 nanomaterials-13-01163-f004:**
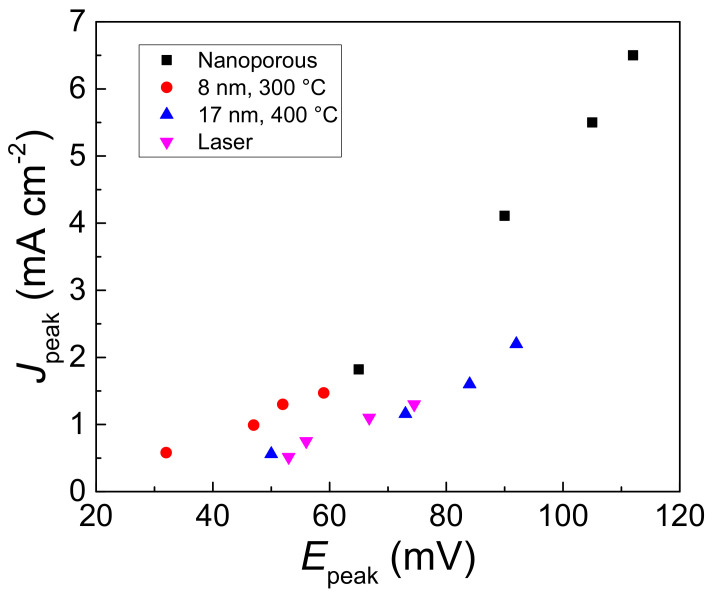
Current density as function of potential of the two electrons oxidation peak of the voltammograms of Figure 3a–d for 2.5; 5; 7.5; and 10 mM glucose in 0.1 M NaOH.

**Figure 5 nanomaterials-13-01163-f005:**
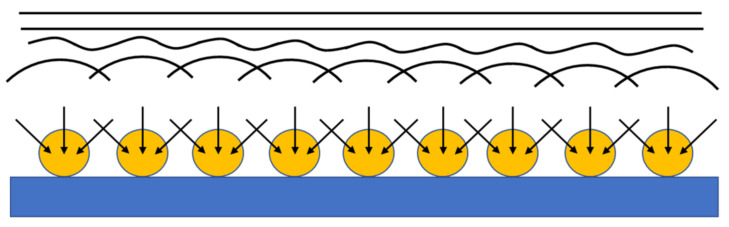
Scheme of modification of convergent diffusion into semi-infinite linear diffusion, when the nanostructure size and interspacing are much smaller than the root mean square of the diffusion length (Equation (11)).

**Figure 6 nanomaterials-13-01163-f006:**
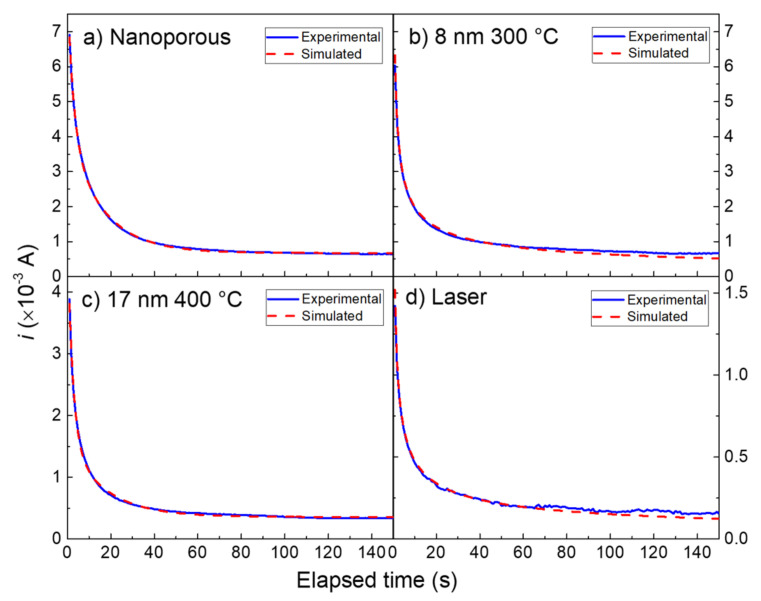
Current-time curves recorded for: (**a**) gold nano-porous; (**b**) 8 nm dewetted at 300 °C; (**c**) 17 nm dewetted at 400 °C; (**d**) laser dewetted. Conditions: glucose 10 mM in NaOH 0.1 M.

**Figure 7 nanomaterials-13-01163-f007:**
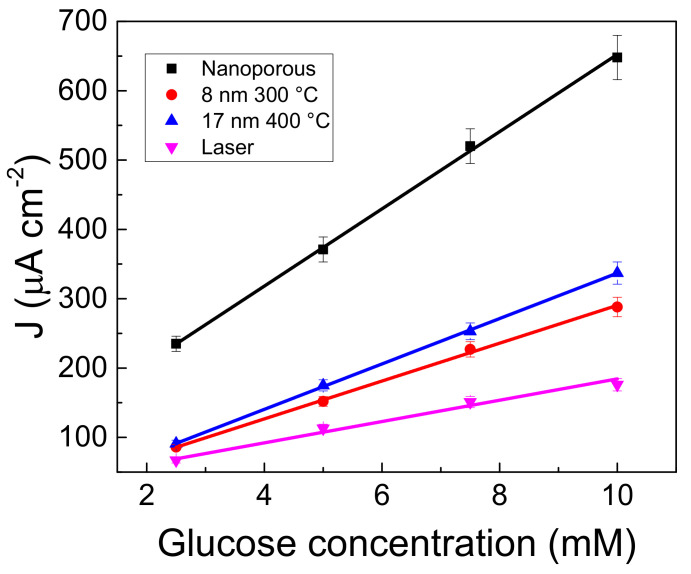
Calibration curves for the glucose determination in amperometric mode. Condition: NaOH 0.1 M.

**Table 1 nanomaterials-13-01163-t001:** Parameters used for the simulation of current-time curves by Cottrel and two-phase exponential decay function.

	Cottrel	Two-Phase Exponential Decay					Adj. R^2^
	B_0_ (A)	$N\;\frac{4nFADc}{\pi r}\;B_{1}\left(A\right)$	$N\;\frac{4nFADc}{\pi r}\;B_{2}\;\left(A\right)$	$N\;\frac{4nFADc}{\pi r}\;B_{3}\;\left(A\right)$	k_1_ (s^−1^)	k_2_ (s^−1^)	
Nanostructure type	-	-	-	-	-	-	-
Nano- porous	-	6.74 × 10^−4^ ± 3.02 × 10^−6^	0.00339	0.00333	0.363	0.0625	0.99924
8 nm 300 °C	0.00633 ± 3.5 × 10^−5^	-	-	-	-	-	0.98530
17 nm 400 °C	-	3.58 × 10^−4^ ± 2.83 × 10^−6^	0.00265	0.00119	0.425	0.0586	0.99715
Laser	0.00152 ± 9.3 × 10^−6^	-	-	-	-	-	0.98172

**Table 2 nanomaterials-13-01163-t002:** parameters used for the linear fit of the calibration curves reported in Figure 7.

Sample	y=a+bx		Adj R^2^
	a (μAcm^−2^)	b (μAcm^−2^mM^−1^)	
Nano-porous	95.49 ± 4.25	55.69 ± 0.89	0.9992
8 nm 300 °C	9.48 ± 1.65	32.74 ± 0.36	0.9996
17 nm 400 °C	17.54 ± 2.58	27.29 ± 0.58	0.9986
Laser	30.60 ± 6.45	15.38 ± 1.24	0.9807

## Data Availability

The detailed data of the study are available from the corresponding authors by request.

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
