# Peer review of "Model of Chronoamperometric Response towards Glucose Sensing by Arrays of Gold Nanostructures Obtained by Laser, Thermal and Wet Processes"

_nanomaterials, 2023, doi:10.3390/nano13071163_

Round 1

Reviewer 1 Report

The submitted manuscript describes the modification of gold sputtered layers deposited on graphene paper by chemical, thermal, and laser treatment. The properties of the obtained materials were tested in order to use them as materials for the development of glucose-sensitive electrodes. The part of the work devoted to the analytical applications of the produced material is very short and limited to presenting only the calibration curves.

There are some issues that need clarification, as follows.

-          What was the working surface of the tested electrodes?

-          In the case of the thermal treatment of gold electrodes, the term ‘annealing’ seems more appropriate than ‘dewetting’.

-          The capture of Figure 1 should provide more information about the materials shown there.

-          ‘In these two gold nanostructures, due to the high surface extension. Does ‘high surface extension’ mean ‘highly developed surface’?

-          What was the stability of the developed materials? Do the authors encounter any problems with adhesion of sputtered gold layers? Was the electrode fabrication procedure repeatable? Was the electrode-to-electrode reproducibility tested? 

Author Response

The submitted manuscript describes the modification of gold sputtered layers deposited on graphene paper by chemical, thermal, and laser treatment. The properties of the obtained materials were tested in order to use them as materials for the development of glucose-sensitive electrodes.

 The part of the work devoted to the analytical applications of the produced material is very short and limited to presenting only the calibration curves.

Author reply: thank you to the reviewer for her/his comment. In this paper, we pay the attention on the modeling of the chronoamperometric response. Analytical applications of our gold nanostructures were thoroughly studied in potentiometric mode and described in previous works. Please vide our previous works:

Scandurra A., Ruffino F., Sanzaro S., Grimaldi M.G., Laser and Thermal Dewetting of Gold Layer onto Graphene Paper for non-Enzymatic Electrochemical Detection of Glucose and Fructose. Sens. Actuators B Chem., 2019, 301, 127113 https://doi.org/10.1016/j.snb.2019.127113;

Scandurra A., Ruffino F., Censabella M., Terrasi A., Grimaldi M. G., Dewetted Gold Nanostructures onto Exfoliated Gra-phene Paper as High Efficient Glucose Sensor. Nanomaterials 2019, 9, 1794; https://doi.org/10.3390/nano9121794

Thus, further description of the analytical applications is beyond the scope of this work. However, to cope with this reviewer’s comment, we add a new sentence at lines 344-345 that clarify this point.

There are some issues that need clarification, as follows.

-          What was the working surface of the tested electrodes?

Author reply: the working surface area is 1 cm2.

-          In the case of the thermal treatment of gold electrodes, the term ‘annealing’ seems more appropriate than ‘dewetting’.

Author reply: we kindly disagree with the reviewer on this point. The annealing of a thin film does not necessarily produce its dewetting. We have deliberately stressed this point because dewetting is what allows us to produce nanoelectrode arrays. For these reasons, we prefer to leave the term dewetting and use the term annealing in some circumstances. In fact, to cope with this reviewer’s comment we specified “dewetting by thermal annealing” in the experimental section at lines 112-115. Moreover, we rewrote more clearly other sentences in the manuscript regarding the laser dewetting.

-          The capture of Figure 1 should provide more information about the materials shown there.

Author reply: to cope with this reviewer’s comment we rewrote the caption of Figure 1.

-      ‘In these two gold nanostructures, due to the high surface extension. Does ‘high surface extension’ mean ‘highly developed surface’?

Author reply: yes, high surface extension means highly developed surface. To cope with this reviewer comment we changed a little the sentence at line 322.

-          What was the stability of the developed materials? Do the authors encounter any problems with adhesion of sputtered gold layers? Was the electrode fabrication procedure repeatable? Was the electrode-to-electrode reproducibility tested?

Author reply: we would like to thank the reviewer for her/his comment. The gold adhesion onto graphene paper is excellent. Also the electrodes stability and repeatability are excellent since the electrodes can be used up to 50 time. The analytical applications were thoroughly studied and described in previous work in potentiometric mode. Please vide:

Scandurra A., Ruffino F., Sanzaro S., Grimaldi M.G., Laser and Thermal Dewetting of Gold Layer onto Graphene Paper for non-Enzymatic Electrochemical Detection of Glucose and Fructose. Sens. Actuators B Chem., 2019, 301, 127113 https://doi.org/10.1016/j.snb.2019.127113;

Scandurra A., Ruffino F., Censabella M., Terrasi A., Grimaldi M. G., Dewetted Gold Nanostructures onto Exfoliated Gra-phene Paper as High Efficient Glucose Sensor. Nanomaterials 2019, 9, 1794; https://doi.org/10.3390/nano9121794

The numbering of equations and references were properly corrected.

Reviewer 2 Report

The manuscript entitled “nanomaterials-2297704” dealing with laser processing has been reviewed. The paper has been nicely written but needs significant improvement. Please follow my comments.

1.     What is the main issue that will be solved by this investigation? Please clarify it in the text.

2.     Please add a brief statement on your methodology in the abstract.

3.     Figure 2 “Au 4f XPS”. How binding energy was calculated. Please add some detail.

4.     What is the future direction of this work?

5.     Please proofread the paper.

6.     Add more detail about the reported values to the conclusion. This increases the bonding of this section to the previous sections and improves the quality of your paper.

7.     Laser has many advantages over the conventional manufacturing method which can be highlighted in your paper. Please read the following manuscript and add it to the literature to show how the laser is comparable with conventional manufacturing.

·       Laser subtractive and laser powder bed fusion of metals: review of process and production features

Author Response

The manuscript entitled “nanomaterials-2297704” dealing with laser processing has been reviewed. The paper has been nicely written but needs significant improvement. Please follow my comments.

  1. What is the main issue that will be solved by this investigation? Please clarify it in the text.

Author reply: the issues that can be solved by our investigation are both of fundamental research as well as for the electrode design. To cope with this reviewer’s comment, we added a new sentence in the conclusion section: “The work we have presented shows how by simulating the amperometric response it is possible to obtain information both on the diffusion model, whether convergent or planar, but also on the mechanisms and kinetics of the electrochemical reactions taking place at the electrode-interface of the solution.”

  1. Please add a brief statement on your methodology in the abstract.

Author reply: to cope with this reviewer’s comment we add a new sentence in the abstract at lines 24-26.

  1. Figure 2 “Au 4f XPS”. How binding energy was calculated. Please add some detail.

Author reply: binding energy scale was calibrated to Au 4f 7/2 of not treated gold film, that is fixed at 84 eV. The method of BE scale calibration was already reported in the Experimental section at line 127. Now the sentence is written more clearly.

  1. What is the future direction of this work?

Author reply: the future direction this work is the application of the simulation method to other nanomaterial-based electrodes employed in the determination of other analyte.

  1. Please proofread the paper.

Author reply: Thank you for this suggestion. We careful checked the manuscript for misprinting and other errors. Now the text is significant improved.

  1. Add more detail about the reported values to the conclusion. This increases the bonding of this section to the previous sections and improves the quality of your paper.

Author reply: thank you for this suggestion. We add a new sentence in the Conclusions section to cope this comment (please vide reply to question 1).

  1. Laser has many advantages over the conventional manufacturing method which can be highlighted in your paper. Please read the following manuscript and add it to the literature to show how the laser is comparable with conventional manufacturing.

Laser subtractive and laser powder bed fusion of metals: review of process and production features

Author reply: we add the reference suggested by the reviewer: Khorasani M., Gibson I., Ghasemi A.H., Hadavi E., Rolfe B. "Laser subtractive and laser powder bed fusion of metals: review of process and production features", Rapid Prototyp. J., 2023, Vol. ahead-of-print No. ahead-of-print. Published online https://doi.org/10.1108/RPJ-03-2021-0055 in the Introduction section and in the References list (new reference # 17).

The numbering of equations and references were properly corrected.

Round 2

Reviewer 2 Report

The paper is ready to publish.